# Workplace Violence and Its Effects on Burnout and Secondary Traumatic Stress among Mental Healthcare Nurses in Japan

**DOI:** 10.3390/ijerph17082747

**Published:** 2020-04-16

**Authors:** Yudai Kobayashi, Misari Oe, Tetsuya Ishida, Michiko Matsuoka, Hiromi Chiba, Naohisa Uchimura

**Affiliations:** Department of Neuropsychiatry, Kurume University School of Medicine, Asahi-machi 67, Kurume 830-0011, Japan

**Keywords:** workplace violence, mental healthcare nurses, secondary traumatic stress, burnout, nursing license

## Abstract

Workplace violence (WPV) in healthcare settings has drawn attention for over 20 years, yet few studies have investigated the association between WPV and psychological consequences. Here, we used a cross-sectional design to investigate (1) the 12-month prevalence of workplace violence (WPV), (2) the characteristics of WPV, and (3) the relationship between WPV and burnout/secondary traumatic stress among 599 mental healthcare nurses (including assistant nurses) from eight hospitals. Over 40% of the respondents had experienced WPV within the past 12 months. A multivariate logistic regression analysis indicated that occupation and burnout were each significantly related to WPV. Secondary traumatic stress was not related to WPV. Our results suggest that WPV may be a long-lasting and/or cumulative stressor rather than a brief, extreme horror experience and may reflect specific characteristics of psychological effects in psychiatric wards. A longitudinal study measuring the severity and frequency of WPV, work- and non-work-related stressors, risk factors, and protective factors is needed, as is the development of a program that helps reduce the psychological burden of mental healthcare nurses due to WPV.

## 1. Introduction

Workplace violence (WPV) is defined as any incident in which a person is abused, threatened, or assaulted in circumstances related to their work; this can include verbal abuse and threats as well as physical attacks [1,2]. The International Labor Organization reported that the magnitude of exposure to violence at work depends not only on a person’s occupation but also upon the circumstances named “situations at risk,” including those associated with working alone (e.g., small shops); working with the public (e.g., railway workers); working with valuables (e.g., financial institutions); working with people in distress (e.g., healthcare workers); working in an environment that is increasingly open to violence (e.g., school teachers); working in conditions of special vulnerability (e.g., immigrant workers); working in military and paramilitary organizations; and working in zones of conflict [3].

Workplace violence (WPV) in healthcare settings has been drawing attention for over 20 years [2,4,5,6,7,8] and has been reported in many places, including Europe [9], Asia [10,11,12], the U.S. [13], and the Middle East [14]. According to a recent review, the prevalence of WPV against healthcare workers was higher in Asian and North American countries than that in other countries [15]. Female nurses were reported to be the victims of verbal abuse more often than male nurses, and male nurses were reported to be more commonly the victims of physical abuse [7]. Most of the WPV in healthcare settings occurs in psychiatric departments, emergency services, polyclinics/waiting rooms, and geriatric units [8].

According to a review of WPV in psychiatric wards over the past 20 years, studies of WPV have examined mainly its occurrence rate, risk assessment, and risk management; fewer investigations have assessed the physical and psychological consequences of WPV [2]. WPV may cause not only physical injuries but also psychological impacts, resulting in higher rates of fear or anxiety, anger, insecurity, depression, emotional exhaustion, suicidal thoughts, post-traumatic stress symptoms, guilt, self-blame, and shame [2,8]. The consequences of WPV include decreased job satisfaction, increased intent to leave the organization, and lowered health-related quality of life [2].

WPV and its consequences among nurses in Japan have been described [10,16,17,18,19,20]. Of respondents to previous surveys, 33–47% of the nurses had experienced WPV during the prior 12 months [16,17,18]. The proportion of nurses who experienced physical aggression and verbal abuse was significantly high in psychiatric wards [16]. Another study showed that nurses who had encountered verbal abuse or violence by patients in psychiatry departments had experienced severe psychological impacts such as secondary traumatic stress and low satisfaction with family support [10]. The characteristics of WPV itself in Japan have been evaluated, but few studies have examined the psychological effects of WPV in detail.

Burnout is defined as a syndrome conceptualized as resulting from chronic workplace stress that has not been successfully managed [21]. It is characterized by three dimensions: feelings of energy depletion or exhaustion; increased mental distance from one’s job or feelings of negativism or cynicism related to one’s job; and reduced professional efficacy [21]. In the International Classification of Diseases, 11th revision (ICD-11), the definition is more detailed: “burnout refers specifically to phenomena in the occupational context and should not be applied experiences in other areas of life” [22]. Among healthcare workers, nurses are known to struggle with burnout symptoms the most, and this poses serious consequences for patients, other healthcare professionals, and healthcare organizations [23]. A meta-analysis on burnout in mental healthcare nurses showed that variables such as work overload, work-related stress, professional seniority, male gender, being single, and aggression at work contributed to burnout development [24]. Burnout is regarded as one of the relevant consequences of WPV [2,25]. Associations between WPV and burnout, turnover intention [26,27], and intention to leave [28] have been described.

Secondary traumatic stress (STS) is a syndrome including intrusion, avoidance, and arousal resulting from indirect traumatic exposure in a professional context [29]. A study conducted in Israel showed that psychiatric nurses reported higher levels of STS symptoms compared to community nurses [30]. Another study revealed a high correlation between work-related post-traumatic stress disorder (PTSD) symptoms (due to traumatic stressors such as WPV), STS, and burnout in psychiatric nurses [31].

We conducted the present study to investigate the prevalence of WPV and its effects on burnout and STS among mental healthcare nurses in Japan. We focused on mental healthcare nurses because there have been few investigations of the association between WPV and psychological consequences. We conducted a large-scale, multicenter study to clarify the psychological impact of WPV by using questionnaires about the respondents’ well-being, psychological distress, alcohol use disorder, and anger related to harmful experiences, as well as compassion satisfaction, burnout and STS. We tested our hypothesis that mental healthcare nurses who have experienced WPV have a higher rate of burnout and higher secondary traumatic stress.

## 2. Participants and Methods 

### 2.1. Study Design

This is a questionnaire-based cross-sectional study.

### 2.2. Participants and Procedures

Mental health nurses and mental health assistant nurses working at the mental health ward of a university hospital or one of seven mental health hospitals (i.e., psychiatric inpatient, outpatient, and day-treatment centers) on the island of Kyushu in Japan were recruited using the convenience (i.e., not randomized) sampling method. The inclusion criterion was age 20–79 years old. Candidates who could not understand the questionnaires in Japanese were excluded. 

The researcher in charge (Y.K.) explained the purpose and methods of the research to the head nurse of the psychiatric neurology ward of A University Hospital (pseudonym) and the head of each hospital both in a letter of request and orally. After obtaining the hospital’s consent to participate in the research in a letter of consent, the researcher in charge distributed a set of questionnaires to the person in charge of the target hospital (the person in charge of the facility at the head of each ward or the secretary-level depending on the hospital). The researcher in charge verbally explained the freedom of research participation, significance, the purpose of use, method of use, questionnaire items, the protection of personal information, the questionnaire handling after collection, and contact information, etc. to the person in charge at the facility. That person then verbally explained the study to the research subjects, i.e., nurses. Participants were encouraged to fill out the questionnaires when they were alone outside of work hours. The questionnaires were hand-delivered by a person in charge at each hospital, and the enclosed envelopes were sealed up to 2 weeks after distribution. The completed questionnaires were dropped into collection boxes placed temporarily at each hospital. The researcher in charge retrieved the questionnaires.

The respondents’ data were collected from October 18 to November 30, 2019. The questionnaires were administered and completed anonymously. We collected sociodemographic data such as sex, age, marital status, occupation, years of experience as a registered nurse or as an assistant nurse, the type of present workplace (acute ward or others). Of 650 eligible nurses, 599 nurses and assistant nurses participated in the study (response rate, 92.2%). We did not obtain the sociodemographic information of the non-respondents.

### 2.3. Measures

#### 2.3.1. Well-Being

The World Health Organization-Five Well-Being Index (WHO-5) is a five-item self-report questionnaire developed by the WHO; it measures the respondent’s current mental well-being [32]. The total raw score, ranging from 0 to 25, is multiplied by 4 to give the final score, with 0 representing the worst imaginable well-being and 100 representing the best imaginable well-being. The Japanese version of the WHO-5 was validated by Awata et al. [33,34] and showed sufficient reliability and validity in community-dwelling elderly persons and in diabetic patients. A cutoff score of the Japanese WHO-5 version at 11/12 or 12/13 of the raw total score for detecting depression was recommended by Awata et al. for the Japanese version [33]. In the present study, we used 12/13 as the cutoff score for low well-being according to Awata’s recommendation. Cronbach’s alpha of the present study was 0.89.

#### 2.3.2. Psychological Distress

The Kessler six-item scale (K6) is a self-report questionnaire measuring psychological distress; it consists of six brief questions [35]. The total score (ranging from 0 to 24) has been used as an indicator of serious mental illness or mood and anxiety disorders in the general population. The Japanese version of the K6 showed screening performance that was essentially equivalent to that of the original English version [36,37]. We adopted the 12/13 cut-off according to Kessler’s recommendation [35]. The Cronbach’s alpha of the present study was 0.88.

#### 2.3.3. Alcohol Use Disorder

The Alcohol Use Disorder Identification Test (AUDIT), which was developed by the WHO for the screening of excessive drinking, is a 10-item self-report questionnaire [38]. The Japanese version of the AUDIT was validated by Kawada et al. [39] and showed satisfactory internal reliability. The score ranges from 0 to 40. In Japan, a total score ≥11 is considered indicative of alcohol abuse [40]. The Cronbach’s alpha of the present study was 0.79.

#### 2.3.4. Anger Related to Harmful Experience

The Dimensions of Anger Reaction-5 (DAR-5) is a five-item self-report questionnaire regarding the respondent’s anger reaction after a harmful experience such as violence. The DAR-5 was developed and validated by Forbes et al. [41,42]. The score ranges from 5 to 25. Because the Japanese version of the DAR-5 has not been developed, we used a back-translated version provided by a principal researcher in this study. According to Forbes et al., the cutoff point of 12 was used for high risk. The Cronbach’s alpha of the present study was 0.80.

#### 2.3.5. Workplace Violence

For assessing WPV, we asked eight questions that were used in a prior study’s questionnaire regarding WPV [43]; that questionnaire was derived from the “WPV in the Health Sector Country Case Studies Research Instruments Survey Questionnaires” [44] as set out by an International Labor Office/International Council of Nurses/WHO/Public Services International (ILO/ICN/WHO/PSI) project. Because there was no Japanese version for these questions, we used a back-translated version provided by a principal researcher. Among the eight questions, four were qualitative yes/no questions (the existence of violence within the past 12 months, the existence of physical injury, whether the respondent thinks that this violence was avoidable, and the presence of help-seeking behavior); one was a quantitative question (psychological influence), and the remaining three questions were qualitative multiple-choice questions (characteristics of violence, perpetrator, type of coping behaviors).

#### 2.3.6. Burnout, Secondary Traumatic Stress, and Compassion Satisfaction

We used the ProQOL (Professional Quality of Life) scale to measure burnout and secondary traumatic stress simultaneously. As a 30-item self-report questionnaire developed by Stamm [45], the ProQOL consists of three subscales: compassion satisfaction, burnout, and secondary traumatic stress. “Compassion satisfaction” is about the pleasure that a person derives from being able to do his/her work well. Burnout and secondary traumatic stress are negative aspects of one’s professional quality of life. In the ProQOL scale, burnout is indicated by exhaustion, frustration, anger, and depressive symptoms. An example of the scale’s burnout subscale questions is “I feel connected to others.” Secondary traumatic stress is about work-related secondary exposure to people who have experienced extremely or traumatically stressful events. The negative effects of secondary traumatic stress can include fear, sleep difficulties, intrusive images, and avoiding reminders of the person’s traumatic experiences. An example of the ProQOL scale’s secondary traumatic stress question is “I jump or am startled by unexpected sounds.”

The Japanese version of the ProQOL was validated by Fukumori et al. [46]. According to those authors, a high risk of compassion satisfaction is indicated by a score of ≤26 points, a high risk of burnout by a score of ≥33 points, and a high risk of secondary traumatic stress by ≥28 points [46]. The Cronbach’s alpha of the present study was 0.78. We considered the fact that burnout and secondary traumatic stress were clearly separated as subscales to be an advantage in the present study.

### 2.4. Statistical Analyses

All statistical analyses were conducted using JMP Pro for Windows, ver. 14 (SAS Institute, Cary, NC, USA). The chi-square (χ^2^) test was used for testing relationships between categorical variables. The Shapiro–Wilk test was used to check normal distributions of the continuous variables. Only the compassion satisfaction subscale of the ProQOL showed a normal distribution. In light of these results, we decided to use non-parametric analyses. The Mann–Whitney U test was used for comparisons of two independent groups. We performed a multivariate logistic regression analysis to test the association between a categorical dependent variable and a set of independent variables. All tests were two-sided and based on a 0.05 level of significance. Pairwise deletion was conducted for missing data.

### 2.5. Ethical Considerations

All subjects gave their informed consent for inclusion before they participated in the study. The study was conducted in accord with the Declaration of Helsinki, and the protocol was approved by the Ethical Committee of Kurume University, Japan (approval no. 19100, approved on 9 September 2019).

## 3. Results

### 3.1. Sociodemographic Characteristics of the Participants

The sociodemographic characteristics of the study participants are summarized in Table 1. Females accounted for 65.4% of the participants, and the mean age was 47.1 years. Regarding the specific nursing occupations, 66.4% of the participants were certified nurses, and 32.6% had an assistant nurse certification. Concerning marital status, 353 (58.9%) of the participants were married, 122 (20.4%) participants were single, 87 (14.5%) participants were divorced, and 30 (5.0%) participants were widowed (missing data; *n* = 7). There were 258 (43.6%) participants working at acute wards, and the others were working at chronic wards or at an outpatient division.

### 3.2. Prevalence of WPV within the Past 12 Months and the Characteristics of the WPV

Among these 599 mental health nurses (including assistant nurses), 265 participants (44.7%) answered that they had experienced WPV within the past 12 months. Regarding the types of violence (multiple choices were available), verbal violence was reported by 165 participants (62.3% of 265 victims); physical violence was reported by 160 participants (60.4%); and sexual violence was reported by seven participants (2.6%). Sex did not affect the percentages of the types of violence. The mean age among the nurses who experienced physical violence was significantly lower than that of the nurses who had not had this experience: (mean ± SD) 44.7 ± 12.0 vs. 48.2 ± 13.4, |z| = 2.9, *p* < 0.01). The percentages of verbal violence and sex violence were not significantly different among the types of working wards, but the percentage of physical violence in the participants working at acute wards (33.3%) was significantly higher than that of the participants working at other places (22.0%, χ^2^ = 9.4, *p* < 0.01). The percentage of sexual violence did not differ by occupation, but there was a significantly higher percentage of verbal violence among the nurses (31.1%) than among the assistant nurses (21.4%, χ^2^ = 6.1, *p* = 0.01), and a significantly higher percentage of physical violence among the nurses (30.8%) than among the assistant nurses (18.2%, χ^2^ = 10.5, *p* < 0.01).

The perpetrators of WPV were the subject of a multiple-choice question: 223 participants (84.2% of 265 victims) reported that the perpetrator was a patient, 11 participants (4.2%) described a patient’s family member, 35 participants (13.2%) reported a work colleague, and 33 participants (12.5%) responded that their boss or supervisor was the perpetrator of WPV.

### 3.3. Mental Health Status of Nurses Who Experience WPV

The mean values and standard deviations of the mental health status results are summarized in Table 1. The number of participants who attained a low value of well-being as the WHO-5 raw score was 310 (52.6%) participants. The number of participants with high psychological distress on the K6 was 42 (7.1%). Fifty-six (10.1%) participants reported problem drinking, and 41 (7.2%) participants were at high risk of anger related to harmful experiences. The number of participants at high risk of compassion satisfaction was 332 (59.4%); that of burnout was 62 (10.9%); and that of secondary traumatic stress was 18 (3.2%).

### 3.4. Comparisons of the Groups with and without Workplace Violence

The results of our comparisons of mental health status between the participants with (*n* = 265) and without (*n* = 328) WPV (missing data; *n* = 6) are provided in Table 1. The χ^2^ analyses revealed that there were significant differences between these two groups in occupation and type of workplace (wards). The nurses had experienced WPV more than the assistant nurses had, and the nurses who were working at acute wards had experienced WPV more compared to the nurses who worked at other wards. There was no significant difference in the experience of WPV based on marital status. The Mann–Whitney U-tests showed that there were significant differences in age, the well-being score, psychological distress, anger, and three subscales of professional quality of life. The participants who experienced WPV were younger and showed lower well-being, higher psychological distress, higher anger, lower compassion satisfaction, higher burnout, and higher secondary traumatic stress.

### 3.5. Risk Factors of Workplace Violence

To investigate the risk factors of WPV, we conducted a multivariate logistic regression analysis using sociodemographic characteristic variables (sex, age, years of experience, occupation, and ward type) and symptom measures (well-being, psychological distress, alcohol use, anger, (low) compassion satisfaction, burnout, and secondary traumatic stress) as listed in Table 2. Among them, occupation and burnout were each identified as a significant factor related to WPV. The odds ratio of nurses experiencing WPV compared to the assistant nurses was 2.03.

## 4. Discussion

### 4.1. Main Findings

The respondents’ questionnaire results demonstrated that >40% of the respondents had experienced WPV within the prior 12 months. Compared to the nurses who did not experience WPV, the respondents who experienced WPV were younger and showed lower well-being scores, higher psychological distress, higher anger, lower compassion satisfaction, higher burnout, and higher secondary traumatic stress. These results revealed that WPV affects the mental health status of staff. The high prevalence of WPV observed in this study population is consistent with those of studies conducted in Japan [16] and other countries [43,47]. The effects of WPV on the present nurses’ mental health status are also in line with those of previous investigations [2,8].

However, our present findings do not support our hypothesis that WPV is associated with higher burnout and higher secondary traumatic stress. In the multivariate regression analysis, only burnout was associated with WPV; secondary traumatic stress was not. The association we observed between WPV and burnout is in agreement with previous studies [25,48,49]. A cross-sectional study conducted in the Netherlands revealed that physical aggression was positively associated with the staff’s burnout symptoms [48], and this association remained in the same study sample over a two-year longitudinal study [49]. The advantage of the present investigation is that we examined burnout and secondary traumatic stress using the ProQOL scale, which can avoid overlapping of symptoms. Our finding that WPV was not associated with secondary traumatic stress suggests that the psychological burden of mental healthcare nurses that is due to WPV may be recognized as a long-lasting and/or cumulative stressor rather than as a brief, extreme horror experience, and it may reflect specific characteristics of psychological effects in psychiatric wards.

Researchers in Israel investigating mental healthcare nurses reported that the association between WPV and burnout/secondary traumatic stress was observed only indirectly via general work stress in the prior month [50]. Those authors explained that mental healthcare nurses may feel that WPV is an integral component of their job, and low burnout/secondary traumatic stress may be an adjusted reaction to their work environment [50,51]. Our present data showing a direct association between WPV and burnout may call this idea into question, and it may suggest that mental healthcare nurses with WPV feel a psychological overload as maladaptation to their work environment.

Interestingly, we also observed that the type of nursing license was a risk factor for WPV; the questionnaire results demonstrated that the nurses experienced more WPV than the assistant nurses. This is not consistent with a recent Japanese study investigating job-related stress [52]. In that study, assistant nurses who were working at psychiatry hospitals experienced more irritability and somatic symptoms than the nurses working there. The authors of that study speculated that stronger stress reactions of assistant nurses were observed because of their lesser education period compared to the certified nurses. In our study, we asked about the existence of WPV, which is categorized into an objective variable. The reason for the higher frequency of WPV among nurses may be because nurses tend to play a more responsible role; for example, persuading a patient to accept medication or a medical treatment or conducting forced treatment. A U.S. study investigating work-related violence based on the type of nursing license (between registered nurses and licensed practical nurses) showed that the risk of physical assault was increased for the licensed practical nurses working with neonatal/pediatric patients, whereas the registered nurses’ risk was decreased; the registered nurses’ risk of physical violence increased while providing patient care, whereas the licensed practical nurses’ risk increased while supervising patient care [53].

Although we did not find an association in multivariate analysis, our present study revealed that psychiatric nurses working at acute wards had encountered more physical violence compared to that in other workplaces for nurses. This is consistent with a study that showed that patients with mental illness (affected by dementia, mental retardation, drug and substance abuse, or other psychiatric disorders) were the most frequent perpetrators of physical violence in a general hospital [54]. It is also of note that verbal violence was observed equally in acute wards and in other places including chronic wards in the present study.

### 4.2. Recommendations for Further Research

We raise four major challenges for further research. First, because we did not measure the severity and frequency of WPV, a study measuring them may play a role. Second, we detected an association between WPV and burnout, but we did not ask the participants about their stressors other than WPV. Moreover, our study was cross-sectional. It is thus necessary to conduct a large-scale longitudinal cohort study to assess the impacts of WPV, work-related stressors, and non-work-related stressors. It has been reported that both WPV and burnout were positively associated with turnover intention [26]. Third, not only risk factors but also factors that protect against negative psychological impacts after WPV should be examined. A study in Canada investigated the role of relational occupational coping self-efficacy against workplace incivility and burnout and revealed that self-coping is a protective factor [55]. Fourth, a program that helps reduce the psychological burden of mental healthcare nurses due to WPV should be developed. A study demonstrated that the life satisfaction of mental healthcare nurses was affected more by staff resilience, post-traumatic growth, and job stress than by WPV [56]. A program with strategies for increasing occupational coping self-efficacy, resilience, and post-traumatic growth may be effective.

### 4.3. Study Limitations

There are several limitations to be considered in this study. (1) The study’s design was cross-sectional, and we cannot discuss causality. (2) We did not include a comparison group such as nurses working in an internal medicine ward. (3) We did not use validated versions of the evaluations for anger and WPV. (4) We did not obtain the sociodemographic information of the non-respondents; this may have caused selection bias.

## 5. Conclusions

The results of our study demonstrated that mental healthcare nurses who experienced WPV showed a poorer mental health status, and WPV was associated with burnout among mental healthcare nurses in Japan. Further research is merited, including longitudinal studies, investigations assessing risk and protective factors, and the development of an intervention program for the psychological burden experienced after WPV. An expansion of the existing call to action by the implementation of policy changes is also important to mitigate workplace violence in healthcare settings.

## Figures and Tables

**Table 1 ijerph-17-02747-t001:** Sociodemographics and comparison of the groups with and without workplace violence within the past 12 months (*n* = 599).

Variables	Total (*n* = 599)	With WPV (*n* = 265)	Without WPV (*n* = 328)	Test	Probability
	*n*	*n* (%)	*n* (%)	χ^2^	*p*
**Sex**					
Female	390	169 (43.7)	218 (56.3)		
Male	206	95 (46.3)	110 (53.7)	0.4	0.5
Missing	3				
**Occupation**					
Nurse	398	197 (49.5)	201 (50.5)		
Assistant nurse	195	64 (33.5)	127 (66.5)	13.6 *	<0.01
Missing	6				
**Ward**					
Acute ward	258	129 (50.2)	128 (49.8)		
Others	334	134 (40.5)	197 (59.5)	5.5 *	0.02
Missing	7				
	**Mean (SD)**	**Mean (SD)**	**Mean (SD)**	**|z|**	***p***
Age, years	47.1 (13.2)	45.7 (11.9)	48.4 (14.0)	2.5 *	0.01
Duration as nurse, years	19.8 (13.4)	19.0 (11.9)	20.5 (14.5)	0.8	0.4
WHO-5 final score	48.2 (19.0)	45.5 (18.7)	50.3 (19.0)	2.8 *	<0.01
K6	5.4 (4.5)	6.2 (4.9)	4.7 (4.1)	3.8*	<0.01
AUDIT	3.9 (4.7)	3.9 (4.8)	3.8 (4.6)	0.4	0.7
DAR-5	7.3 (2.6)	7.9 (3.0)	6.8 (2.2)	5.2 *	<0.01
Compassion satisfaction	24.4 (8.7)	23.5 (9.0)	25.2 (8.4)	2.3*	0.02
Burnout	25.4 (6.4)	26.5 (5.8)	24.4 (6.8)	3.3*	<0.01
Secondary traumatic stress	12.0 (6.8)	13.6 (7.1)	10.6 (6.2)	5.4*	<0.01

WPV: workplace violence; WHO-5: World Health Organization-Five Well-Being Index; K6: Kessler Six-Item Scale; AUDIT: Alcohol Use Disorder Identification Test; DAR-5: Dimensions of Anger Reaction-5. * *p* < 0.05.

**Table 2 ijerph-17-02747-t002:** Multivariate logistic regression analysis for workplace violence within the past 12 months.

Independent Variable	Beta	Wald	*p*	OR	95% CI
Sex, 1: male, 0: female	−0.08	0.57	0.45	0.85	0.55–1.31
Age	−0.01	0.70	0.40	0.99	0.96–1.01
Years of experience	−0.005	0.16	0.69	1.00	0.97–1.01
Occupation, 1: Nurse, 0: Assistant nurse	0.35	9.96	<0.01	2.03	1.31–3.15
Ward, 1: Acute ward, 0: other	0.09	0.72	0.39	1.19	0.80–1.78
WHO-5, 1: raw score ≤12, 0: others	0.04	0.17	0.68	1.09	0.72–1.67
K6, 1: ≥13, 0: others	0.28	1.70	0.19	1.75	0.75–4.06
AUDIT, 1: ≥11, 0: others	0.13	0.53	0.47	1.29	0.65–2.55
DAR-5, 1: ≥12, 0: others	0.29	2.09	0.15	1.78	0.81–3.90
Compassion satisfaction, 1: ≤26, 0: others	0.05	0.24	0.62	1.11	0.73–1.70
Burnout, 1: ≥33, 0: others	0.34	4.33	0.04	1.99	1.04–3.81
Secondary traumatic stress, 1: ≥28, 0: others	0.22	0.46	0.50	1.55	0.44–5.50

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
