# Peer review of "Workplace Violence and Its Effects on Burnout and Secondary Traumatic Stress among Mental Healthcare Nurses in Japan"

_ijerph, 2020, doi:10.3390/ijerph17082747_

Round 1

Reviewer 1 Report

Thank you for the opportunity to review this manuscript. Please see comments below for specific feedback.

  1. Workplace violence is defined within the existing literature many different ways  a definition should be included within the introduction section.
  2. Line 44 states only a single article is available that describes the relationship between WPB and ProQOL. I was able to identify several others please see citations below and revise as appropriate.

    Alshehry, Abdualrahman Saeed. “Workplace Incivility and Its Influence on Professional Quality of Life among Nurses from Multicultural Background: A Crosssectional Study.” Journal of Clinical Nursing, vol. 28, no. 13/14, 2019, pp. 2553–2564., doi:https://doi-org.ezproxy-v.musc.edu/10.1111/jocn.14840.

    Choi, Seung 10Hye. “Workplace Violence against Nurses in Korea and Its Impact on Professional Quality of Life and Turnover Intention.” Journal of Nursing Management, vol. 25, no. 7, 2017, pp. 508–518., doi:https://doi-org.ezproxy-v.musc.edu/10.1111/jonm.12488.

  3. Incorporation of a conceptual framework and/or theory to guide your results would be helpful. 

Author Response

Thank you for your valuable comments. Here are our responses.

1 Workplace violence is defined within the existing literature many different ways a definition should be included within the introduction section.

Response: Thank you for your helpful comments. In accord with your advice, we added the definition of workplace violence in the Introduction section.

2 Line 44 states only a single article is available that describes the relationship between WPB and ProQOL. I was able to identify several others please see citations below and revise as appropriate. 

Response: Because we changed the framework of this manuscript, we do not use the word "professional QOL" as a concept. We focus on the associations between WPV and burnout and secondary traumatic stress. We revised the entire manuscript and have read the above-cited studies and now cite them as references.

3 Incorporation of a conceptual framework and/or theory to guide your results would be helpful.

Response: In accord with your advice, we re-considered the conceptual framework of this study and focused on the associations between WPV and burnout and secondary traumatic stress. We thus revised part of the Results section. We newly observed that occupation and burnout were significantly related to WPV and that secondary traumatic stress was not related to WPV. We revised the Discussion section in accord with our new findings.

Reviewer 2 Report

Comment: can you better explain the sample recruitment methodology? Are there any criteria for inclusion or exclusion of the sample? Do you have information (demographic characteristics) on non-responders?

Author Response

Thank you for your valuable comments. Here are our responses.

Comment: can you better explain the sample recruitment methodology? Are there any criteria for inclusion or exclusion of the sample? Do you have information (demographic characteristics) on non-responders?

Response: Thank you for your valuable comments. In accord with your advice, we added a description of the sample recruitment methodology including the inclusion and exclusion criteria of the participant sample in the Methods section. We did not obtain the sociodemographic information of the non-respondents. We added this information to the Methods and Discussion (limitations) sections.

Reviewer 3 Report

I want to thank you for the opportunity to review this manuscript

The theme is interesting and there is a need to investigate this aspect with accurate scientific reports.

The paper is overall sufficiently clear and well written. Some concerns have to be solved before re-evaluating the manuscript and possibly considering it for publication.

In details:

Abstract: change your abstract accordingly with the concerns reported below.

Introduction: nowadays, the theme of the workplace violence in health setting is no so recent and new, as you reported in the abstract and in the introduction. I suggest to improve literature research also including studies from the past 10 years and most international recent studies about the theme (some explicative paper are sufficient). You reported global data about the WPV, but what about the data about the country where the study was performed?

 I recommend delving deeper into the theme of the proQOL. This is a crucial part of the introduction and it needs a strong improvement. You must report studies about antecedents and consequences of a low proQOL. What organizational factors can affect proQOL? What can be the consequences of a low proQOL. Why there is a need to investigate this theme? The scientific gap is overall very weak. State clearly what was the motivation to perform this kind of study and what was the hypothesis you want to test with the present study. All these information are needed to justify your study setting. Overall, introduction need a strong improvement. Following the STROBE guidelines for observational studies, in the introduction you must “ Explain the scientific background and rationale for the investigation being reported” and “State specific objectives, including any prespecified hypotheses”. Both these aspects are substantially lacking in your introduction.

Participants and methods

Study population has to be properly described. Population recruitment, and a description of the population characteristics are strongly recommended. How did you involve the participants? How did you select them? Inclusion and exclusion criteria? Where the study was conducted? How did you trait missing values?

Study design and study setting are totally absent. Following the STROBE guidelines for observational studies you must

  • Present key elements of study design early in the paper
  • Describe the setting, locations, and relevant dates, including periods of recruitment, exposure, follow-up, and data collection

Who administered the questionnaires? Are they administered during working time? What kind of personal data did you collect? Did you consider the job tenure of the workers?

I suggest to add a subheading “procedures” to explain all these concepts.

Measures:

Well-being: you correctly described the procedure to obtain the final score, starting from the row score. But you considered a row score cut-off to classify the population. Why did you consider the row score instead of the final score you calculated? Moreover, why did you chose 12/13 as cut off? Every decision has to be scientifically motivated. You can not decide in an arbitrary way. The cut-off is so crucial that a wrong choice can strongly influence the results. No Cronbach alpha for the well-being index?

Psychological distress: who validated the Japanese K6 version?

“Cronbach's alpha of the present study was…”. The alpha refers to the subscale, not to the study in general.

Ethical considerations: when the study has been approved?

Discussion:

Try to explain how WPV affect mental health status. I think this relationship is quite obvious. What is new in your results? Are they similar to other investigation about the theme? References and comparisons are mainly based on local study. This aspect can compromise the external validity of the results. You have to consider also international literature about the theme. 

Interpretation  and generalisability of the results are lacking in your discussion, and they need a strong improvement.

It is not clear what this paper adds to the scientific community, and how your results can contribute to fill a gap in the scientific literature. At this stage it seems a well conducted study without any add value. Try to enhance your interesting results.

Conclusion: “Our study clearly demonstrated that mental health nurses who experienced WPV showed a  poorer mental health status than those without experiences of WPV”. You can not omit the results of the multivariate analysis that does not confirm the hypothesis. So, conclusions are not supported by the results.

Best regards

Author Response

Thank you for your helpful comments. Here are our responses.

Abstract: change your abstract accordingly with the concerns reported below.

Response: Thank you for your helpful comments. We revised the Abstract in accord with your advice.

Introduction: nowadays, the theme of the workplace violence in health setting is no so recent and new, as you reported in the abstract and in the introduction. I suggest to improve literature research also including studies from the past 10 years and most international recent studies about the theme (some explicative paper are sufficient). You reported global data about the WPV, but what about the data about the country where the study was performed?

Response: We completely agree with your comment and have revised the Introduction section accordingly. We read several reviews on workplace violence and now introduce them in the Introduction section. We also added data on WPV in Japan, where this study was performed.

I recommend delving deeper into the theme of the proQOL. This is a crucial part of the introduction and it needs a strong improvement. You must report studies about antecedents and consequences of a low proQOL. What organizational factors can affect proQOL? What can be the consequences of a low proQOL. Why there is a need to investigate this theme? The scientific gap is overall very weak. State clearly what was the motivation to perform this kind of study and what was the hypothesis you want to test with the present study. All these information are needed to justify your study setting. Overall, introduction need a strong improvement. Following the STROBE guidelines for observational studies, in the introduction you must “ Explain the scientific background and rationale for the investigation being reported” and “State specific objectives, including any prespecified hypotheses”. Both these aspects are substantially lacking in your introduction.

Response: After careful reconsideration, we decided not to use the term "profession QOL" as a concept, because it is difficult to interpret the concept of compassion satisfaction (which is a positive aspect) and to understand the overall picture of professional QOL. Therefore, we focused only on burnout and secondary traumatic stress in the latest version of the manuscript. The term "ProQOL" is used only as the name of a scale. We also changed variables in the multivariate analysis to focus on the association between WPV and burnout/secondary traumatic stress. In the previous version of the manuscript, we had set each subscale of the ProQOL scale as dependent variables. Doing so might have caused confusion or misunderstanding. In the latest version of the manuscript, we set WPV as the dependent variable in the logistic regression analysis.

Participants and methods

Study population has to be properly described. Population recruitment, and a description of the population characteristics are strongly recommended. How did you involve the participants? How did you select them? Inclusion and exclusion criteria? Where the study was conducted? How did you trait missing values?

Response: We added information on the population recruitment, population characteristics, sampling method, inclusion and exclusion criteria, and the sites of the study to the "Procedure" section. Our sampling was based on the convenience (i.e., not randomized) sampling method. The inclusion criterion was age 20–79 years old, and the exclusion criterion was being unable to understand/complete the questionnaires in Japanese. Pairwise deletion was conducted for missing data. This information was added to the Statistical Analysis section.

Study design and study setting are totally absent. Following the STROBE guidelines for observational studies you must

  • Present key elements of study design early in the paper
  • Describe the setting, locations, and relevant dates, including periods of recruitment, exposure, follow-up, and data collection

Who administered the questionnaires? Are they administered during working time? What kind of personal data did you collect? Did you consider the job tenure of the workers?

I suggest to add a subheading “procedures” to explain all these concepts.

Response: In accord with your advice, we added a description of the study design and study setting information in the Procedures section. We collected sociodemographic data such as sex, age, marital status, occupation, years of experience as a registered nurse or assistant nurse, and the type of present workplace (acute ward or others). We considered the years of experience of the nurses and included it as a variable in all analyses.

Measures:

Well-being: you correctly described the procedure to obtain the final score, starting from the row score. But you considered a row score cut-off to classify the population. Why did you consider the row score instead of the final score you calculated? Moreover, why did you chose 12/13 as cut off? Every decision has to be scientifically motivated. You can not decide in an arbitrary way. The cut-off is so crucial that a wrong choice can strongly influence the results. No Cronbach alpha for the well-being index?

Response: Cut-off score setting in the Japanese version was examined by Awata et al. (Awata S, Bech P, Yoshida S et al. Reliability and validity of the Japanese version of the World Health Organization-Five Well-Being Index in the context of detecting depression in diabetic patients. Psychiatry Clin Neurosci 2007, 61, 112-119, doi:10.1111/j.1440-1819.2007.01619.x). They had used raw scores, and we followed their definition. We added this information to the Measurements section. Cronbach's alpha was 0.89, and this information had been presented in the middle of the original paragraph. We moved that sentence to the end of the paragraph.

Psychological distress: who validated the Japanese K6 version?

Response: Furukawa et al. validated the Japanese version of K6. We added this information to the Measurements section.

“Cronbach's alpha of the present study was…”. The alpha refers to the subscale, not to the study in general.

Response: We think that our description in the previous version of the manuscript was confusing because we had written that "it consists of six brief questions about depression and anxiety". Because the K6 has no subscales (six items for the whole K6), we revised the sentence as "The Kessler six-item scale (K6) is a self-report questionnaire measuring psychological distress; it consists of six brief questions."

Ethical considerations: when the study has been approved?

Response: The protocol was approved on September 9, 2019. We added this information to the Statistical Analyses section.

Discussion:

Try to explain how WPV affect mental health status. I think this relationship is quite obvious. What is new in your results? Are they similar to other investigation about the theme? References and comparisons are mainly based on local study. This aspect can compromise the external validity of the results. You have to consider also international literature about the theme.

Response: We detected a significant association between WPV and burnout. We revised the Discussion section in accord with your advice. We have now cited both domestic and international studies.

Interpretation and generalisability of the results are lacking in your discussion, and they need a strong improvement.

Response: We added our interpretation and generalizability in the Discussion section as follows:

"Our finding that WPV was not associated with secondary traumatic stress suggests that the psychological burden of mental healthcare nurses that is due to WPV may be recognized as a long-lasting and/or cumulative stressor rather than a brief, extreme horror experience, and it may reflect specific characteristics of psychological effects in psychiatric wards."

It is not clear what this paper adds to the scientific community, and how your results can contribute to fill a gap in the scientific literature. At this stage it seems a well conducted study without any add value. Try to enhance your interesting results.

Response: Please see our response above.

Conclusion: “Our study clearly demonstrated that mental health nurses who experienced WPV showed a poorer mental health status than those without experiences of WPV”. You can not omit the results of the multivariate analysis that does not confirm the hypothesis. So, conclusions are not supported by the results.

Response: We changed the description of our conclusion as follows:

"The results of our study demonstrated that mental healthcare nurses who experienced WPV showed a poorer mental health status, and WPV was associated with burnout among mental healthcare nurses in Japan. Further research is merited, including longitudinal studies, investigations assessing risk and protective factors, and the development of an intervention program for the psychological burden experienced after WPV. An expansion of the existing call to action by the implementation of policy changes is also important to mitigate workplace violence in healthcare settings."

Reviewer 4 Report

Since workplace violence is both unreported and underreported, research suggests increased awareness is requisite to formulating control strategies (Firenze et al., 2020). This study contributes to existing research on workplace violence in Italy (Firenze et al., 2020), India (Hossain et al., 2020), China (Shi et al., 2020), Taiwan (Lee et al., 2020), and PubMed literature (Mento et al., 2020).  By correlating workplace violence with nurses’ mental health, burnout, secondary traumatic stress (compassion fatigue), and quality of life, the authors validate the need for more effective workplace violence prevention measures.

Manuscript strengths include:

  • Definitive quantitative data creates evidence-based research
  • Correlations between workplace violence, anger, burnout, and compassion fatigue validate the magnitude of this issue
  • Study sheds light on growing global challenge:  "WHO recently declared burnout as a “occupational phenomenon” in the International Classification of Diseases 11th revision (ICD-11), recognizing burnout as a serious health issue. Amongst healthcare workers, nurses are known to struggle with burnout symptoms the most, carrying serious consequences for patients, other healthcare professionals and healthcare organisations."  Source:  Woo, T., Ho, R., Tang, A., & Tam, W. (2020). Global prevalence of burnout symptoms among nurses: A systematic review and meta-analysis. Journal of Psychiatric Research123, 9-20.

Opportunities for improvement include:

  • Discuss inconsistent, contradictory results in a separate section
  • Create "recommendations for further research" to clarify areas meriting additional attention to better understand nuances of these phenomena
  • Expand existing call to action by formulating possible intervention studies and policy changes to mitigate workplace violence in healthcare settings

References

Firenze, A., Santangelo, O. E., Gianfredi, V., Alagna, E., Cedrone, F., Provenzano, S., & La Torre, G. (2020). Violence on doctors. An observational study in Northern Italy. La Medicina del Lavoro,111(1), 46-53.

Hossain, M. M., Sharma, R., Tasnim, S., Al Kibria, G. M., Sultana, A., & Saxena, T. (2020). Prevalence, characteristics, and associated factors of workplace violence against healthcare professionals in India: A systematic review and meta analysis. medRxiv. https://doi.org/10.1101/2020.01.01.20016295

Lee, H. L., Han, C. Y., Redley, B., Lin, C. C., Lee, M. Y., & Chang, W. (2020). Workplace violence against emergency nurses in Taiwan: a cross-sectional study. Journal of Emergency Nursing,46(1), 66-71.

Mento, C., Silvestri, M. C., Bruno, A., Muscatello, M. R. A., Cedro, C., Pandolfo, G., & Zoccali, R. A. (2020). Workplace violence against healthcare professionals: A systematic review. Aggression and Violent Behavior,51, 101381.

Shi, L., Li, G., Hao, J., Wang, W., Chen, W., Liu, S., ... & Zhang, L. (2020). Psychological depletion in physicians and nurses exposed to workplace violence: A cross-sectional study using propensity score analysis. International Journal of Nursing Studies, 103, 103493.

Author Response

Thank you for your valuable comments. Here are our responses.

Since workplace violence is both unreported and underreported, research suggests increased awareness is requisite to formulating control strategies (Firenze et al., 2020). This study contributes to existing research on workplace violence in Italy (Firenze et al., 2020), India (Hossain et al., 2020), China (Shi et al., 2020), Taiwan (Lee et al., 2020), and PubMed literature (Mento et al., 2020). By correlating workplace violence with nurses’ mental health, burnout, secondary traumatic stress (compassion fatigue), and quality of life, the authors validate the need for more effective workplace violence prevention measures.

Response: Thank you for your helpful comments. We added most of the references that you mentioned in the introduction section. One study in India was excluded because that article has not been published.

Manuscript strengths include:

  • Definitive quantitative data creates evidence-based research
  • Correlations between workplace violence, anger, burnout, and compassion fatigue validate the magnitude of this issue
  • Study sheds light on growing global challenge:  "WHO recently declared burnout as a “occupational phenomenon” in the International Classification of Diseases 11th revision (ICD-11), recognizing burnout as a serious health issue. Amongst healthcare workers, nurses are known to struggle with burnout symptoms the most, carrying serious consequences for patients, other healthcare professionals and healthcare organisations."  Source:  Woo, T., Ho, R., Tang, A., & Tam, W. (2020). Global prevalence of burnout symptoms among nurses: A systematic review and meta-analysis. Journal of Psychiatric Research123, 9-20.

Response: We added your description on burnout and the reference in the Introduction section.

Opportunities for improvement include:

  • Discuss inconsistent, contradictory results in a separate section
  • Create "recommendations for further research" to clarify areas meriting additional attention to better understand nuances of these phenomena
  • Expand existing call to action by formulating possible intervention studies and policy changes to mitigate workplace violence in healthcare settings

Response: We reconsidered the conceptual framework of this study and focused on the associations between WPV and burnout and secondary traumatic stress and revised the manuscript accordingly. We created a "Recommendations for further research" section and clarified areas meriting additional attention to gain a better understanding of the nuances of these phenomena. We revised the conclusion as follows:

"The results of our study demonstrated that mental healthcare nurses who experienced WPV showed a poorer mental health status, and WPV was associated with burnout among mental healthcare nurses in Japan. Further research is merited, including longitudinal studies, investigations assessing risk and protective factors, and the development of an intervention program for the psychological burden experienced after WPV. An expansion of the existing call to action by the implementation of policy changes is also important to mitigate workplace violence in healthcare settings."

Round 2

Reviewer 3 Report

change 2.2 "Procedures" in "Participants and Procedures" 

Author Response

Thank you for your comment. In accord with your advice, we changed "2.2 Procedures" into "2.2 Participants and Procedures".